# Multi-Label Open Set Recognition

**Yi-Bo Wang, Jun-Yi Hang, Min-Ling Zhang**[*]
School of Computer Science and Engineering, Southeast University, Nanjing 210096, China
Key Laboratory of Computer Network and Information Integration (Southeast University),
Ministry of Education, China
{wang_yb, hangjy, zhangml}@seu.edu.cn

## Abstract

In multi-label learning, each training instance is associated with multiple labels simultaneously. Traditional multi-label learning studies primarily focus on closed set scenario, i.e. the class label set of test data is identical to those used in training phase. Nevertheless, in numerous real-world scenarios, the environment is open and dynamic where unknown labels may emerge gradually during testing. In this paper, the problem of multi-label open set recognition (MLOSR) is investigated, which poses significant challenges in classifying and recognizing instances with unknown labels in multi-label setting. To enable open set multi-label prediction, a novel approach named SLAN is proposed by leveraging sub-labeling information enriched by structural information in the feature space. Accordingly, unknown labels are recognized by differentiating the sub-labeling information from holistic supervision. Experimental results on various datasets validate the effectiveness of the proposed approach in dealing with the MLOSR problem.

## 1   Introduction

*Multi-label Learning* (MLL) deals with the problem where an instance can be associated with multiple labels simultaneously [37, 19]. As a practical machine learning paradigm, multi-label learning has been widely applied in various real-world applications, such as image annotation [30], text categorization [26], information retrieval [12].

Traditional multi-label learning studies focus on closed set scenario. That is, they assume that the class label set of test data is identical to that in the training set [37, 8, 21]. However, in many real-world scenarios, this assumption rarely holds because the environment is open and dynamic [29]. In addition to the extant label knowledge at training phase, the unknown labels may emerge gradually with the data streams during the testing phase. For example, in Figure 1, the test image is annotated with several relevant labels, some of which are unseen in the training set. The classification task becomes much more challenging because the label correlation between known and unknown labels may degrade the performance of the predictive model. Furthermore, due to the presence of unknown labels in the class label set of test data, these test data are hardly employed in subsequent learning processes, such as incremental learning [15].

Motivated by the potential applications, we formalize a novel framework named *multi-label open set recognition* (MLOSR), whose goal is to learn a multi-label model that can correctly classify known labels for the unseen instance and recognize unknown labels within its relevant label set. This can be regarded as a special weakly supervised learning. In MLOSR, the most challenging part is to recognize the unknown labels associated with instances. Since we do not have any prior knowledge of unknown labels and they almost always co-occur with some known labels, it is very difficult to separate instances with unknown labels from those with the known labels only.

---

[*]Corresponding author

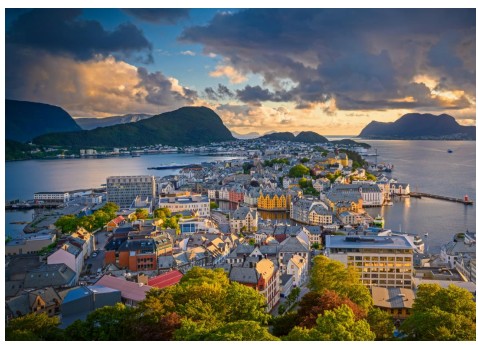

**The relevant label set**

cloud      building

car      tree

ship      sky

sea      trestle bridge

mountain    ...

Figure 1: An example. The test image is associated with a variety of relevant labels. Among the set of relevant labels, "cloud", "car", "ship", "building", "tree" and "sky" are known labels seen in the training set, while "sea", "mountain" and "trestle bridge" are unknown labels emerging in the testing phase.

*Open set recognition* (OSR) is a paradigm previously proposed in [24] and is formalized as a risk-minimizing constrained functional optimization problem. OSR describes a scenario where new classes unseen in training occur in testing, thus classifiers must be able to properly identify seen samples while rejecting unseen ones [9]. [24] proposes a 1-vs-Set machine to minimize open set risk by sculpting a decision space from the marginal distances of binary SVM. OSR problem is further studied via algorithm adaptation [1, 16], statistical extreme value theory (EVT) [25, 32], margin distribution [23] or hierarchical Dirichlet process [10]. These existing works are based on the fact that each instance owns one ground-truth label for multi-class cases. Thus, they cannot be used to directly solve the MLOSR problem, due to the multiple ground-truth labels in MLL.

To address MLOSR problem, we propose a tailored algorithm named SLAN, i.e. *Sub-Labeling informAtion reconstructioN for multi-label open set recognition*. The basic strategy of SLAN is to enrich the sub-labeling information in the sub-label space by leveraging the structural information in the feature space and differentiating it from the labeling information from holistic supervision. Specifically, the underlying structure of feature space is characterized by the sparse reconstruction relationships among training instances. After that, the reconstruction information is utilized to guide the enrichment of sub-labeling information. Then, a unified optimization framework is presented to simultaneously facilitate open set recognizer and multi-label classifier induced with alternating optimization. Our empirical study on datasets from diverse domains demonstrates the effectiveness of the proposed approach.

The rest of the paper is organized as follows. We present a brief review of related works. Then we formulate the problem and propose the algorithm. Next, experimental results are reported, followed by the conclusion.

## 2   Related Work

The task of multi-label learning has been extensively studied in recent years [37, 19]. Generally, the major challenge for multi-label learning is its huge output space which is exponential to the number of class labels. Therefore, exploiting *label correlations* has been adopted as a common strategy to facilitate the learning process. Roughly speaking, extant approaches can be grouped into three categories based on the order of correlations, i.e. *first-order* approaches, *second-order* approaches and *high-order* approaches. First-order approaches tackle multi-label learning problem in a label-by-label manner [2, 33]. Second-order approaches exploit pairwise interactions among class labels [7, 3]. High-order approaches exploit relationships among a subset of or all class labels [22, 14].

OSR is critical for the tasks where incomplete knowledge exists at training time, and unknown classes can be submitted to an algorithm during the testing phase, requiring the classifiers to classify the seen classes and deal with unseen ones. According to [9], traditional machine learning methods are adapted to OSR scenario. For instance, SVM-based models add extra constraints on the score space in [24, 4]; A collective decision-based model implemented by hierarchical Dirichlet process is proposed in [10]; and distance-based models [1, 16] are developed by modifying existing classifiers, such as nearest class mean classifier and nearest neighbor classifier. Some other approaches focus

on the EVT [17]. [25] combines the EVT for score calibration with two separated compact abating probability SVMs, where the first SVM is used as a conditioner and the second SVM fitted yields the posterior estimate. [32] transforms the OSR problem into a set of hypothesis testing problems by modeling the tail part of reconstruction error distribution via EVT. [23] formulates the extreme value machine with distributional information, which can be interpreted by EVT. There are also some methods trying to incorporate few-shot learning into OSR [5].

MLOSR can be regarded as a combination of MLL and OSR. Thus, a straightforward approach is to generate a independent recognizer besides the multi-label classifier. However, as unknown labels may co-occur with known labels, it is difficult to separate instances with unknown labels from instances with known labels only, which leads to OSR approaches that could not be applied in MLOSR problems.

Streaming multi-label learning (SMLL) [31] is similar to our MLOSR problem but differs in the setting of unknown labels. It aims to derive a unified model by taking care of the continually emerging new unknown labels on the training data. [31] trains a linear classifier for new labels with the linear hypotheses between labels and classifiers. [28] proposes a novel DNN-based framework to model the emerging new labels depending on high-order representations. [29] presents probabilistic streaming label tree to incorporate new labels, which capture hierarchical correlations among labels. Compared to SMLL, MLOSR is much more challenging in recognizing unknown labels as the unknown labels only emerge in testing phase. In the next section, the first attempt towards MLOSR is introduced.

## 3 The SLAN Approach

### 3.1 Problem Formulation

Formally, let $\mathcal{X} = \mathbb{R}^d$ denote the $d$-dimensional input space and $\mathcal{Y} = \{l_1, l_2, \ldots, l_q\}$ denote the label space including $q$ class labels. Each multi-label instance can be denoted as $(\boldsymbol{x}_i, Y_i)$, where $\boldsymbol{x}_i \in \mathcal{X}$ is its feature vector and $Y_i \subseteq \mathcal{Y}$ is the set of relevant labels associated with $\boldsymbol{x}_i$. Here, a $q$-dimensional indicator vector $\boldsymbol{y}_i = [y_{i1}, y_{i2}, ...y_{iq}]^\top$ is utilized to denote $Y_i$, where $y_{ik} = 1$ indicates class label $l_k \in Y$ and $y_{ik} = -1$ otherwise. By arranging feature vectors and label vectors of $m$ training instances, we obtain the feature matrix $\mathbf{X} = [\boldsymbol{x}_1, \ldots, \boldsymbol{x}_m]$ and label matrix $\mathbf{Y} = [\boldsymbol{y}_1, \ldots, \boldsymbol{y}_m]$.

Given the multi-label training set $\mathcal{D} = \{(\boldsymbol{x}_i, Y_i) \mid 1 \leq i \leq m\}$, the goal of MLOSR is to learn a model from $\mathcal{D}$ that can correctly classify known labels for the unseen instance and recognize unknown labels within its relevant label set. Conceptually, given the multi-label training set $\mathcal{D}$, an open space risk function $R_{\mathcal{O}}$ and an empirical risk function $R_\varepsilon$, multi-label open set recognition aims to derive a measurable recognition function $f \in \mathcal{H}$ by minimizing the following **Open Set Risk**:

$$\operatorname*{argmin}_{f \in \mathcal{H}} R_{\mathcal{O}}(f) + \lambda_r R_\varepsilon(f(\mathcal{D})) \tag{1}$$

where $\lambda_r$ is a regularization constant.

### 3.2 Structural Information Discovery

To characterize the underlying manifold structure of feature space, a weighted directed graph $G = (\mathcal{V}, \mathcal{E}, \mathbf{S})$ is constructed over the set of training instances, where $\mathcal{V} = \{\boldsymbol{x}_i \mid 0 \leq i \leq m\}$ corresponds to the set of vertices and $\mathcal{E} = \{(\boldsymbol{x}_i, \boldsymbol{x}_j) \mid s_{ij} \neq 0, 1 \leq i \neq j \leq m\}$ corresponds to the set of edges from $\boldsymbol{x}_i$ to $\boldsymbol{x}_j$ with nonzero weight.

Furthermore, $\mathbf{S} = [s_{ij}]_{m \times m}$ corresponds to the weight matrix encoding the relationships among all training instances. Conceptually, the weight value $s_{ij}$ reflects relative importance of $\boldsymbol{x}_i$ in reconstructing $\boldsymbol{x}_j$. Thus, by implementing global sparse reconstruction, the weight matrix $\mathbf{S}$ is instantiated by solving the following optimization problem:

$$\min_{\mathbf{S}} ||\mathbf{XS} - \mathbf{X}||_{\mathrm{F}}^2 + \mu_0 ||\mathbf{S}||_1$$
$$\text{s.t.} \quad s_{ii} = 0, \forall 1 \leq i \leq m \tag{2}$$

Here, the first term controls the linear reconstruction error via squared Frobenius norm while the second term controls the sparsity of reconstruction via $\ell_1$ norm. The relative importance is balanced

by the trade-off parameter $\mu_0$. Then, the constrained optimization problem in Eq.(2) can be solved via a standard ADMM (Alternating Direction Method of Multiplier) procedure [11].

## 3.3 Sub-Labeling Information Enrichment

According to the manifold assumption, the structural relationship specified in the feature space should also be preserved in the entire label space to enrich labeling information originally encoding in the indicator vector $\boldsymbol{y}_i$ [13, 38]. That is, $\boldsymbol{y}_i$ can be transformed into a numerical labeling vector $\boldsymbol{z}_i = [z_{i1}, z_{i2}, \ldots, z_{iq}]^\top$ under holistic supervision which encodes richer semantics for predictive model induction.

However, this assumption might be suboptimal in the entire label space. Considering one specific label $l_k \in \mathcal{Y}$, let $\boldsymbol{f}_i^k = [f_{i1}^k, \ldots, f_{i,k-1}^k, f_{i,k+1}^k, \ldots, f_{iq}^k]^\top$ represent the enriched sub-labeling information of $\boldsymbol{x}_i$. With the manifold assumption, the structural information would be maintained in the sub-label space $\mathcal{Y} \backslash \{l_k\}$. Then, instances with $l_k$ can be reconstructed via the instances without $l_k$ as well as the enriched sub-labeling information. Nevertheless, these reconstructed instances can not be assigned with $l_k$ since there's not enough positive labeling information w.r.t. $l_k$. That is, the sub-labeling information is differentiated from labeling information with holistic supervision. Thus, in open set scenario, if $l_k$ is specified as an unknown class label, such difference can be employed as a criterion for one specific instance to distinguish whether it is associated with an unknown class label.

Let concatenate all $\boldsymbol{f}_i^k$, denoted by $\mathbf{F}_k = [\boldsymbol{f}_1^k, \ldots, \boldsymbol{f}_m^k] \in \mathbb{R}^{(q-1) \times m}$ and all $\boldsymbol{z}_i$, denoted by $\mathbf{Z} = [\boldsymbol{z}_1, \ldots, \boldsymbol{z}_m] \in \mathbb{R}^{q \times m}$. The enriched sub-labeling information is generated via leveraging the structural information encoded in $\mathbf{S}$ by solving the following optimization problem:

$$
\min_{\substack{\mathbf{Z}, \\ \mathbf{F}_1, \ldots, \mathbf{F}_q}} \frac{\gamma}{2} \sum_{i=1}^m \|\boldsymbol{z}_i - \boldsymbol{y}_i\|_2^2 + \frac{\beta}{2} \sum_{k=1}^q \sum_{i=1}^m \|\boldsymbol{f}_i^k - \sum_{j=1}^m s_{ji} \boldsymbol{f}_j^k\|_2^2
$$
$$
+ \frac{\alpha}{2} \sum_{k=1}^q \sum_{i=1}^m \|\delta_i^k (\boldsymbol{f}_i^k - \mathbf{P}_k \boldsymbol{z}_i)\|_2^2 \tag{3}
$$

Here, $\delta_i^k$ is a indicator variable, where $\delta_i^k = 1$ if $l_k$ is not associated with $\boldsymbol{x}_i$; otherwise $\delta_i^k = 0$. $\mathbf{P}_k$ is a $(q-1) \times q$ projection matrix to align $\boldsymbol{z}_i$ with $\boldsymbol{f}_i^k$ without regard to $l_k$, which removes the $k$-th row of the identity matrix. The first term supervises the labeling information from the holistic aspect. The second term conveys the manifold structure specified in the feature space to the sub-label space. The third term minimizes the labeling information difference between sub-label space and entire label space to differentiate instances with or without such unknown class label, which implicitly reduces the open space risk.

During the testing phase, for an unseen instance $\boldsymbol{x}_*$, the reconstruction coefficients w.r.t. $l_k$ is identified by resorting to the ADMM technique over the training set. After that, the enriched sub-labeling information $\boldsymbol{f}_*^k$ of $\boldsymbol{x}_*$ is determined via the second term of Eq.(3). Thereafter, whether unknown labels are associated with $\boldsymbol{x}_*$ is determined by the following recognizer $G$ via ensemble majority voting:

$$
G(\boldsymbol{x}_*) = \begin{cases} \text{unknown}, & if \ \sum_{k=1}^q g_k(\boldsymbol{x}_*) \le 0, \\ \text{known}, & if \ \sum_{k=1}^q g_k(\boldsymbol{x}_*) > 0. \end{cases} \tag{4}
$$

where $g_k(\boldsymbol{x}_*) = -1$ if $\|\boldsymbol{f}_*^k - \mathbf{P}_k \boldsymbol{z}_*\|_2^2 > \rho_k$; otherwise $g_k(\boldsymbol{x}_*) = 1$. $\rho_k$ is the threshold, and can be chosen so that $100 \times \tau\%$ instances with $l_k$ in training set satisfy $\|\boldsymbol{f}_i^k - \mathbf{P}_k \boldsymbol{z}_i\|_2^2 > \rho_k$. The labeling information $\boldsymbol{z}_*$ under holistic supervision is generated by the following MLL classifier.

## 3.4 MLL Classifier Training

Similar to the existing strategy in previous OSR algorithms, instead of training the classifier independently, we perform open set recognizer and multi-label classifier induced simultaneously. Then, the sub-labeling information can be optimized by considering both the manifold assumption and model outputs. Furthermore, considering the label correlation between known and unknown labels, such jointly optimization procedure with recognizer can facilitate the classifier more robust. We denote $\mathbf{W} \in \mathbb{R}^{q \times d}$ and $\mathbf{b} \in \mathbb{R}^q$ as a multi-label classifier and adopt the squared Frobenius norm as the

regularization term to control the model complexity:

$$\min_{\boldsymbol{W},\boldsymbol{b}} \sum_{i=1}^{m} \frac{1}{2}||\mathbf{W}\boldsymbol{x}_i + \mathbf{b} - \boldsymbol{z}_i||_2^2 + \frac{\mu_1}{2}||\mathbf{W}||_F^2 \tag{5}$$

Let $\mathbf{B_k} = [b_{pi}^k]_{(q-1)\times m}$ denote the indicator matrix with $b_{pi}^k = \delta_i^k$. Then, the objective function of the unified framework is shown as follows:

$$\min_{\substack{\mathbf{W},\mathbf{b},\mathbf{Z}, \\ \mathbf{F}_1,\ldots,\mathbf{F}_q}} \sum_{k=1}^{q} (\frac{\beta}{2}||\mathbf{F}_k\mathbf{S} - \mathbf{F}_k||_F^2 + \frac{\alpha}{2}||\mathbf{B}_k \circ (\mathbf{P}_k\mathbf{Z} - \mathbf{F}_k)||_F^2)$$

$$+ \frac{1}{2}||\mathbf{Z} - (\mathbf{W}\mathbf{X} + \mathbf{b}\mathbf{1}_n^\top)||_F^2 + \frac{\gamma}{2}||\mathbf{Z} - \mathbf{Y}||_F^2 \tag{6}$$

$$+ \frac{\mu_1}{2}||\mathbf{W}||_F^2$$

Here, The first two terms control the open space risk and remaining terms control the empirical risk.

## 3.5 Alternative Optimization

**Update Z** With fixed $\mathbf{F}_1,\ldots,\mathbf{F}_q$, $\mathbf{W}$ and $\mathbf{b}$, the optimization problem Eq.(6) can be stated as follows:

$$\min_{\mathbf{Z}} \frac{1}{2}||\mathbf{Z} - (\mathbf{W}\mathbf{X} + \mathbf{b}\mathbf{1}_n^\top)||_F^2 + \frac{\gamma}{2}||\mathbf{Z} - \mathbf{Y}||_F^2$$

$$+ \frac{\alpha}{2} \sum_{k=1}^{q} ||\mathbf{B}_k \circ (\mathbf{P}_k\mathbf{Z} - \mathbf{F}_k)||_F^2 \tag{7}$$

The above optimization problem can be solved by updating $\mathbf{Z}$ with gradient descent. The gradient of the objective function w.r.t. $\mathbf{Z}$ is

$$\nabla\mathbf{Z} = (\mathbf{Z} - (\mathbf{W}\mathbf{X} + \mathbf{b}\mathbf{1}_n^\top)) + \gamma(\mathbf{Z} - \mathbf{Y})$$

$$+ \alpha \sum_{k=1}^{q} \mathbf{P}_k^\top (\mathbf{B}_k \circ (\mathbf{P}_k\mathbf{Z} - \mathbf{F}_k)) \tag{8}$$

**Update $\mathbf{F}_1,\ldots,\mathbf{F}_q$** With $\mathbf{Z}$, $\mathbf{W}$ and $\mathbf{b}$ fixed, the optimization problem Eq.(6) can be stated as follows:

$$\min_{\mathbf{F}_k} \frac{\beta}{2}||\mathbf{F}_k\mathbf{S} - \mathbf{F}_k||_F^2 + \frac{\alpha}{2}||\mathbf{B}_k \circ (\mathbf{P}_k\mathbf{Z} - \mathbf{F}_k)||_F^2 \tag{9}$$

Similarly, gradient descent is employed, and the gradient w.r.t. $\mathbf{F}_k$ is:

$$\nabla\mathbf{F}_k = \beta\mathbf{F}_k\mathbf{T} + \alpha\mathbf{B}_k \circ (\mathbf{F}_k - \mathbf{P}_k\mathbf{Z}) \tag{10}$$

where $\mathbf{T} = (\mathbf{S} - \mathbf{I}_{m\times m})(\mathbf{S} - \mathbf{I}_{m\times m})^\top$.

**Update W and b** While $\mathbf{Z}$ and $\mathbf{F}_1,\ldots,\mathbf{F}_q$ are fixed, the optimization problem (4) can be stated as follows:

$$\min_{\mathbf{W},\mathbf{b}} tr(\mathbf{E}\mathbf{E}^\top) + \mu_1 tr(\mathbf{W}\mathbf{W}^\top)$$

$$\text{s.t.} \quad \mathbf{Z} = \mathbf{W}\mathbf{X} + \mathbf{b}\mathbf{1}_m^\top + \mathbf{E} \tag{11}$$

Here, $\mathbf{E} = [\boldsymbol{e}_1,\ldots,\boldsymbol{e}_m] \in \mathbb{R}^{q\times m}$, where $\boldsymbol{e}_i = \boldsymbol{z}_i - (\mathbf{W}\boldsymbol{x}_i + \mathbf{b})$. To achieve better performance of the predictive model, a kernel extension is further facilitated for the general nonlinear case. Let $\boldsymbol{\Phi} = [\phi(\boldsymbol{x}_1),\ldots,\phi(\boldsymbol{x}_m)] \in \mathbb{R}^{h\times m}$, where $\phi(\bullet) : \mathbb{R}^d \to \mathbb{R}^h$ corresponds to the feature mapping that maps the feature space to some higher dimensional Hilbert space with $h$ dimensions. Then, the Lagrangian function of this problem can be formulated as:

$$\mathcal{L}(\mathbf{W},\mathbf{b},\mathbf{E},\mathbf{A}) = tr(\mathbf{E}\mathbf{E}^\top) + \mu_1 tr(\mathbf{W}\mathbf{W}^\top)$$

$$- tr(\mathbf{A}^\top(\mathbf{W}\boldsymbol{\Phi} + \mathbf{b}\mathbf{1}_m^\top + \mathbf{E} - \mathbf{Z})) \tag{12}$$

---

**Algorithm 1** The pseudo-code of SLAN

---

**Input**: The multi-label training set $\mathcal{D}$, the trade-off parameters $\alpha, \beta, \gamma, \mu_1, \tau$, an unseen instance $\boldsymbol{x}_*$;
**Output**: The predicted label set $Y_*$ for $\boldsymbol{x}_*$, the recognition result $G(x_*)$.
**Process**:
 1: Instantiate the weighted graph $G = (\mathcal{V}, \mathcal{E}, \mathbf{S})$ by solving Eq.(2) with ADMM procedure;
 2: Calculate the kernel matrix $\mathbf{K} = [\kappa(\boldsymbol{x}_i, \boldsymbol{x}_j)]_{m \times m}$;
 3: Initialize $\mathbf{Z}$ with $\mathbf{Y}$;
 4: Initialize $\mathbf{F}_k$ with $\mathbf{P}_k \mathbf{Y}$ $(1 \leq k \leq q)$;
 5: **repeat**
 6:     Update $\mathbf{Z}$ according to Eq.(7);
 7:     Update $\mathbf{F}_k$ according to Eq.(9);
 8:     Update $\mathbf{W}$ and $\mathbf{b}$ according to Eq.(11);
 9: **until** convergence or maximum number of iterations being reached
10: **return** $Y_*$ and $G(\boldsymbol{x}_*)$ according to Eq.(14) and Eq.(4).

---

where $\mathbf{A} = [a_{ki}] \in \mathbb{R}^{q \times m}$ stores the Lagrange multipliers. According to the **KKT** conditions, we can obtain:

$$\mathbf{b} = \frac{\mathbf{Z} \mathbf{H}^{-1} \mathbf{1}_m}{\mathbf{1}_m^\top \mathbf{H}^{-1} \mathbf{1}_m}$$

$$\mathbf{A} = (\mathbf{Z} - \mathbf{b} \mathbf{1}_m^\top) \mathbf{H}^{-1} \tag{13}$$

where $\mathbf{H} = \frac{1}{\mu_1} \mathbf{K} + \mathbf{I}_{m \times m}$ and $\mathbf{K} = \mathbf{\Phi}^\top \mathbf{\Phi}$ with its element $k_{ij} = \kappa(\boldsymbol{x}_i, \boldsymbol{x}_j) = \phi(\boldsymbol{x}_i)^\top \phi(\boldsymbol{x}_j)$ based on the chosen kernel function $\kappa(\cdot, \cdot)$. Then, by incorporating the specified kernel function, the modeling output is denoted by $\frac{1}{\mu_1} \mathbf{A} \mathbf{K} + \mathbf{b} \mathbf{1}_m^\top$. Furthermore, given an unseen instance $\boldsymbol{x}_* \in \mathcal{X}$, its relevant label set is predicted as:

$$Y_* = \{l_k \mid \sum_{i=1}^m a_{ki} \kappa(\boldsymbol{x}_*, \boldsymbol{x}_i) \geq 0, 0 \leq k \leq q\} \tag{14}$$

The pseudo-code of SLAN is summarized in Algorithm 1. Given the multi-label training set, a weighted graph is constructed to characterize the manifold structure of feature space (Step 1). After that, the alternative optimization strategy is adopted to optimize open set recognizer and multi-label classifier simultaneously (Step 2-9). Finally, the relevant label set of the unseen instance is predicted and the recognition result is generated based on the learned model (Step 10).

Table 1: Characteristics of experimental data sets.

| Dataset | $|\mathcal{S}|$ | $dim(\mathcal{S})$ | $L(\mathcal{S})$ | $LCard(\mathcal{S})$ | $LDen(\mathcal{S})$ | $DL(\mathcal{S})$ | $PDL(\mathcal{S})$ |
|---------|------|------|-----|-------|-------|------|-------|
| llog | 1208 | 925 | 17 | 0.966 | 0.057 | 96 | 0.079 |
| enron | 1702 | 1001 | 24 | 3.124 | 0.130 | 548 | 0.322 |
| slashdot | 3659 | 1079 | 14 | 1.173 | 0.084 | 119 | 0.033 |
| recreation | 5000 | 606 | 15 | 1.361 | 0.091 | 259 | 0.052 |
| corel5k | 5000 | 499 | 44 | 2.214 | 0.050 | 1037 | 0.207 |
| arts | 5000 | 462 | 14 | 1.512 | 0.108 | 314 | 0.063 |
| education | 5000 | 550 | 11 | 1.374 | 0.125 | 173 | 0.035 |
| rcvsubset2-2 | 6000 | 944 | 39 | 2.170 | 0.056 | 489 | 0.082 |
| bibtex | 7395 | 1835 | 27 | 0.954 | 0.035 | 380 | 0.051 |

## 4 Experiments

### 4.1 Experimental Setup

Table 1 summarizes the detailed characteristics of each benchmark multi-label data set $\mathcal{S}$ employed in the experiments, including the number of instances $|\mathcal{S}|$, number of features $dim(\mathcal{S})$, number of class labels $L(\mathcal{S})$, label cardinality $LCard(\mathcal{S})$, label density $LDen(\mathcal{S})$, number of distinct label sets $DL(\mathcal{S})$ and proportion of distinct label sets $PDL(S)$. To alleviate the influence of extreme

Table 2: Experimental results of each compared approach (mean±std) with different label batch size (denoted by #label). The best and the second best performance of each data set methods are highlighted in **boldface** and underline respectively. In addition, •/○ indicates whether SLAN is statistically superior/inferior to the comparing approaches on each data set with pairwise t-test (at 0.05 significance level).

| Dataset | #label | LIFT | MUENLPLR | SENCE | LIMIC | SLAN |
|---|---|---|---|---|---|---|
| | | | *Ranking loss* (the smaller, the better) | | | |
| llog | 0 | 0.339±0.033 | 0.400±0.025• | **0.335±0.039** | 0.350±0.035 | 0.344±0.030 |
| | 3 | 0.358±0.035 | 0.418±0.026• | **0.357±0.041** | 0.367±0.040 | 0.359±0.033 |
| | 5 | 0.365±0.036 | 0.421±0.025• | **0.362±0.037** | 0.373±0.037 | 0.366±0.033 |
| | 7 | 0.368±0.032 | 0.425±0.022• | **0.366±0.033** | 0.376±0.033 | 0.370±0.030 |
| | 9 | 0.368±0.028 | 0.426±0.019• | **0.367±0.029** | 0.377±0.032 | 0.374±0.030 |
| enron | 0 | 0.174±0.021• | 0.236±0.024• | 0.159±0.018 | 0.199±0.031• | **0.157±0.016** |
| | 6 | 0.179±0.014• | 0.241±0.011• | 0.172±0.009 | 0.194±0.011• | **0.169±0.011** |
| | 9 | 0.179±0.013• | 0.245±0.013• | 0.174±0.008 | 0.193±0.012• | **0.172±0.012** |
| | 12 | 0.180±0.011• | 0.245±0.012• | 0.174±0.005 | 0.193±0.012• | **0.174±0.010** |
| recreation | 0 | 0.237±0.013• | 0.332±0.018• | **0.214±0.020** | 0.279±0.018• | **0.214±0.020** |
| | 4 | 0.258±0.012• | 0.343±0.015• | **0.242±0.021**○ | 0.286±0.017• | 0.247±0.019 |
| | 6 | 0.260±0.013 | 0.346±0.015• | **0.247±0.019** | 0.287±0.018• | 0.252±0.016 |
| | 8 | 0.264±0.014 | 0.347±0.015• | **0.251±0.018**○ | 0.286±0.018• | 0.258±0.014 |
| slashdot | 0 | 0.312±0.026• | 0.406±0.024• | 0.276±0.026• | 0.324±0.028• | **0.249±0.028** |
| | 3 | 0.324±0.022• | 0.412±0.019• | 0.289±0.020• | 0.337±0.029• | **0.260±0.022** |
| | 5 | 0.335±0.019• | 0.418±0.011• | 0.300±0.015• | 0.350±0.025• | **0.268±0.019** |
| | 7 | 0.335±0.021• | 0.418±0.011• | 0.301±0.021• | 0.351±0.024• | **0.270±0.018** |
| corel5k | 0 | 0.206±0.006○ | 0.317±0.011• | **0.195±0.006**○ | 0.267±0.045 | 0.240±0.006 |
| | 7 | 0.231±0.020○ | 0.332±0.018• | **0.221±0.022**○ | 0.287±0.051 | 0.266±0.013 |
| | 12 | 0.239±0.017○ | 0.337±0.019• | **0.231±0.019**○ | 0.292±0.051 | 0.276±0.012 |
| | 17 | 0.247±0.018○ | 0.342±0.016• | **0.238±0.020**○ | 0.295±0.051 | 0.283±0.012 |
| | 22 | 0.252±0.018○ | 0.345±0.012• | **0.243±0.020**○ | 0.299±0.052 | 0.286±0.013 |
| arts | 0 | 0.220±0.024• | 0.305±0.030• | 0.193±0.023• | 0.297±0.045• | **0.187±0.018** |
| | 3 | 0.324±0.049• | 0.376±0.040• | 0.311±0.051• | 0.397±0.076• | **0.296±0.049** |
| | 5 | 0.348±0.029• | 0.392±0.029• | 0.335±0.033• | 0.421±0.079• | **0.318±0.030** |
| | 7 | 0.363±0.036• | 0.409±0.027• | 0.351±0.042• | 0.431±0.071• | **0.332±0.034** |
| education | 0 | 0.222±0.028• | 0.303±0.034• | 0.194±0.023• | 0.275±0.040• | **0.185±0.029** |
| | 4 | 0.300±0.059• | 0.354±0.056• | 0.278±0.050• | 0.344±0.076• | **0.270±0.049** |
| | 6 | 0.311±0.037• | 0.365±0.037• | 0.290±0.032• | 0.356±0.046• | **0.281±0.034** |
| rcvsubset2-2 | 0 | 0.120±0.005• | 0.259±0.012• | **0.093±0.005**○ | 0.226±0.048• | 0.097±0.007 |
| | 8 | 0.220±0.035• | 0.323±0.027• | **0.197±0.033** | 0.279±0.042• | **0.197±0.025** |
| | 12 | 0.235±0.030• | 0.333±0.024• | 0.212±0.028 | 0.290±0.049• | **0.211±0.027** |
| | 16 | 0.248±0.020• | 0.339±0.016• | 0.226±0.020 | 0.298±0.042• | **0.224±0.017** |
| | 20 | 0.263±0.020• | 0.350±0.017• | 0.241±0.021 | 0.309±0.040• | **0.238±0.019** |
| bibtex | 0 | 0.217±0.026• | 0.396±0.009• | 0.249±0.050• | 0.267±0.013• | **0.114±0.016** |
| | 5 | 0.214±0.031• | 0.387±0.013• | 0.245±0.050• | 0.262±0.024• | **0.114±0.017** |
| | 8 | 0.216±0.035• | 0.388±0.017• | 0.247±0.053• | 0.264±0.034• | **0.117±0.017** |
| | 11 | 0.221±0.037• | 0.391±0.016• | 0.250±0.053• | 0.270±0.026• | **0.122±0.020** |
| | 14 | 0.223±0.031• | 0.390±0.010• | 0.252±0.044• | 0.272±0.019• | **0.123±0.016** |

imbalance, any class label with rare appearance or with overly-high imbalance ratio is excluded from the label space following previous settings [34].

For a given dataset, we randomly select 50% labels as known labels and the remaining labels as unknown labels with different label batch sizes. Then, we sample 60% instances without unknown labels to form training set while the remaining instances are treated as test data. The sampling procedure is repeated ten times, and the mean metric value as well as standard deviation for each label batch are reported.

To evaluate the performance of multi-label classifiers, we utilize five widely-used multi-label evaluation metrics [37], including *Ranking loss*, *One-error*, *Coverage*, *Average precision* and *Macro-averaging AUC*. In addition, *F-measure* is employed to evaluate the performance of recognizers.

To validate the effectiveness of the proposed SLAN approach in multi-label learning, four multi-label learning approaches are used for comparative studies.

- LIFT [35]: A multi-label learning approach, which induces classifiers with the label-specific features generated via conducting clustering analysis for each class label. [parameter configuration: $r = 0.1$]

Table 3: Experimental results of each compared approach (mean±std) with different label batch size (denoted by #label). The best and the second best performance of each data set methods are highlighted in **boldface** and underline respectively. In addition, ●/○ indicates whether SLAN is statistically superior/inferior to the comparing approaches on each data set with pairwise t-test (at 0.05 significance level).

| Dataset | #label | OC-SVM | IFOREST | MUENLFOREST | SLAN |
|---|---|---|---|---|---|
| | | *F-measure* (the greater, the better) | | | |
| llog | 3 | 0.501±0.063● | 0.494±0.062● | 0.340±0.067● | **0.672±0.058** |
| | 5 | 0.440±0.064● | 0.436±0.060● | 0.309±0.067● | **0.576±0.062** |
| | 7 | 0.391±0.049● | 0.387±0.046● | 0.281±0.057● | **0.501±0.050** |
| | 9 | 0.362±0.047● | 0.359±0.043● | 0.267±0.054● | **0.456±0.043** |
| enron | 6 | 0.352±0.100● | 0.350±0.099● | 0.398±0.116 | **0.406±0.096** |
| | 9 | 0.283±0.109● | 0.282±0.108● | 0.314±0.116 | **0.321±0.095** |
| | 12 | 0.251±0.113● | 0.250±0.112● | 0.276±0.120 | **0.281±0.098** |
| recreation | 4 | 0.346±0.066● | 0.339±0.066● | 0.318±0.054● | **0.376±0.066** |
| | 6 | 0.283±0.052● | 0.279±0.052● | 0.265±0.044● | **0.305±0.055** |
| | 8 | 0.230±0.037● | 0.227±0.035● | 0.218±0.033● | **0.244±0.039** |
| slashdot | 3 | 0.411±0.100● | 0.409±0.099● | 0.310±0.088● | **0.528±0.127** |
| | 5 | 0.350±0.094● | 0.349±0.095● | 0.274±0.088● | **0.428±0.105** |
| | 7 | 0.293±0.071● | 0.293±0.072● | 0.235±0.075● | **0.350±0.069** |
| corel5k | 7 | 0.496±0.018● | 0.489±0.016● | 0.327±0.013● | **0.653±0.015** |
| | 12 | 0.430±0.022● | 0.424±0.019● | 0.298±0.013● | **0.539±0.017** |
| | 17 | 0.378±0.018● | 0.373±0.018● | 0.272±0.009● | **0.462±0.012** |
| | 22 | 0.344±0.018● | 0.340±0.017● | 0.253±0.009● | **0.414±0.015** |
| arts | 3 | 0.356±0.074● | 0.354±0.069● | 0.340±0.058● | **0.395±0.105** |
| | 5 | 0.273±0.040 | 0.271±0.037 | 0.264±0.033 | **0.288±0.049** |
| | 7 | 0.230±0.044 | 0.229±0.042 | 0.224±0.038 | **0.235±0.047** |
| education | 4 | 0.282±0.086 | **0.282±0.088** | 0.274±0.082 | 0.277±0.059 |
| | 6 | 0.230±0.060 | **0.231±0.060** | 0.226±0.057 | 0.224±0.040 |
| rcvsubset2-2 | 8 | 0.479±0.031● | 0.475±0.031● | 0.387±0.017● | **0.616±0.025** |
| | 12 | 0.418±0.034● | 0.414±0.032● | 0.346±0.021● | **0.518±0.027** |
| | 16 | 0.374±0.020● | 0.370±0.020● | 0.315±0.014● | **0.452±0.013** |
| | 20 | 0.341±0.021● | 0.338±0.022● | 0.291±0.015● | **0.404±0.014** |
| bibtex | 5 | 0.567±0.021● | 0.564±0.020● | 0.478±0.023● | **0.684±0.025** |
| | 8 | 0.526±0.028● | 0.524±0.026● | 0.450±0.027● | **0.621±0.024** |
| | 11 | 0.489±0.026● | 0.487±0.024● | 0.424±0.024● | **0.563±0.028** |
| | 14 | 0.460±0.023● | 0.459±0.022● | 0.403±0.022● | **0.518±0.029** |

- MUENLPLR [39]: A SVM-based dynamic multi-label learning approach which trains a set of linear classifiers by minimizing misclassification loss and pairwise ranking loss. [parameter configuration: $C_1 = 1, C_2 = 1$]

- SENCE [27]: A multi-label learning approach based on label-specific features generated by mixture-based clustering ensemble. [parameter configuration: $r$= 0.4].

- LIMIC [21]: A multi-semantics multi-label metric learning approach coupled with ML-LNN [36] which learns one global and multiple label-specific local metrics simultaneously. [parameter configuration: $\lambda_1 = 1, \lambda_2 = 100, \gamma = 2, K = 10$].

With the first attempt towards solving the MLOSR problem, there is no method can be directly applied. Thus, we compare the proposed SLAN approach with existing anomaly detection approaches.

- OC-SVM [20]: A SVM-based approach which constructs a hyper-sphere surrounding all instances from known labels.

- IFOREST [18]: An unsupervised forest-based anomaly detection approach which employs average path length over all trees as the anomaly score.

- MUENLFOREST [39]: A forest-based dynamic multi-label learning approach which utilizes clustering process in each nodes by considering the feature space and the label patterns. [parameter configuration: $q = 5, \psi = 256, g = 100, e_m = 9$].

For the proposed SLAN approach, trade-off parameters are set as $\alpha = 0.1, \beta = 0.1, \gamma = 10, \mu_1 = 0.1, \tau = 0.8$. $\mu_0$ is fixed to be 0.1. The sensitivity analysis of parameter configurations is conducted in Subsection 4.3. A Linux server equiped with Intel Xeon CPU (48 cores @ 2.67GHz) and 256GB memory is used for supporting the experiments.

## 4.2 Experimental Results

The detailed experimental results in terms of *Ranking loss* and *F-measure* are reported in Table 2-3. Due to the page limit, the results on other metrics are shown in the Appendix. Meanwhile, pairwise t-test [6] is conducted to demonstrate whether the performance of SLAN is statistically superior/inferior to the comparing approaches on each data set. The resulting win/tie/loss counts are summarized in the supplementary material.

Based on the reported experimental results, the following observations can be made:

- Compared with the performance on close set instances (#label = 0), the performance of all comparing approaches on open set instances degrades. Nonetheless, across all multi-label evaluation metrics, SLAN achieves superior or at least comparable performance against the comparing approaches in 91.7% cases. The results clearly indicate the jointly optimization with recognizer serves a more effective way to achieve more robust multi-label classifier in the open environment.

- Comparing with anomaly detection approaches, SLAN achieves better performance in 83.3% cases. Possible reasons are that: (a) OC-SVM and IFOREST are previously designed for multi-class scenario which can not directly solve MLOSR problem. (b) For the construction of MUENLFOREST, instance is augmented with its predictive values derived from MUENLPLR. However, the predictive values might be suboptimal as MUENLPLR is trained without considering open space risk.

- In the multi-label setting, instances with unknown labels may share the same dense region of instances with known labels, which makes multi-class anomaly detection approaches tend to reject instances with unknown labels. That is why since MUENLPLR is under multi-label setting, it still inferior to IFOREST and OC-SVM.

## 4.3 Parameter Sensitivity Analysis

In this section, we study the sensitivity analysis of trade-off parameters $\alpha, \beta, \gamma, \mu_1, \tau$ shown in Algorithm 1. Figure 2 illustrates how the performance of SLAN changes with varying parameter configurations on data set enron. As shown in Figure 2, SLAN achieves relatively stable performance on multi-label metrics and somewhat sensitive on *F-measure*. In this paper, trade-off parameters are set as $\alpha = 0.1, \beta = 0.1, \gamma = 10, \mu_1 = 0.1, \tau = 0.8$, which can be employed as the default parameter setting.

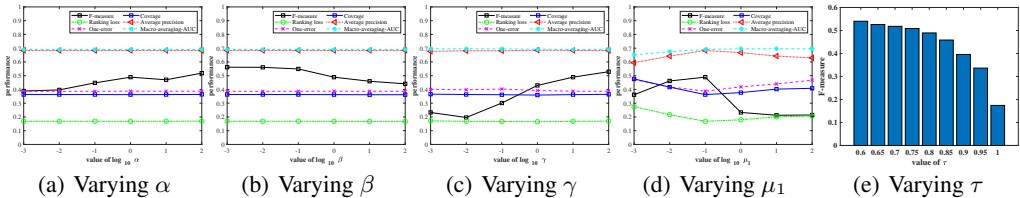

| (a) Varying $\alpha$ | (b) Varying $\beta$ | (c) Varying $\gamma$ | (d) Varying $\mu_1$ | (e) Varying $\tau$ |

Figure 2: Performance of SLAN with varying value of trade-off parameters on enron.

## 5 Conclusion

The major contributions of our work are two-fold: 1) We formalize a novel learning framework named multi-label open set recognition (MLOSR), which aims to classify and recognize instances with unknown labels in multi-label setting, suggesting a new direction for multi-label learning. 2) We propose a novel MLOSR approach named SLAN which can facilitate open set multi-label classification by utilizing sub-labeling information and recognize the unknown labels by differentiating the sub-labeling information from holistic supervision. Extensive experimental results clearly validate the effectiveness of the proposed SLAN approach.

However, SLAN enriches #labels+1 sub-labeling information, which could hardly generalize to extreme multi-label data set. Meanwhile, SLAN works in the multi-label learning schema where

the feature representations of instances may less informative. In the future, it is interesting to investigate towards extreme multi-label learning to achieve tolerable scalability and design deep MLOSR approaches with discriminative feature representations. Furthermore, it is desirable to extend evaluation metrics for MLOSR.

## Acknowledgments

The authors wish to thank the anonymous reviewers for their helpful comments and suggestions. This work was supported by the National Science Foundation of China (62225602) and the Big Data Computing Center of Southeast University.

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

# A Appendix

## A.1 Experimental Results

Table 4, 5, 7 and 9 report detailed experimental results in terms of *Average precision*, *Macro-averaging AUC*, *Coverage* and *One-error*, which are not covered in the *Experimental Results* part of the main body due to page limit. Besides, the pairwise t-test is conducted to demonstrate whether the performance of SLAN is statistically superior/inferior to the comparing approaches on each data set. The resulting win/tie/loss counts in terms of multi-label evaluation metrics are summarized in Table 6 as well as *F-measure* in Table 8.

Table 4: Experimental results of each compared approach (mean±std) with different label batch size (denoted by #label). The best and the second best performance of each data set methods are highlighted in **boldface** and underline respectively. In addition, ●/○ indicates whether SLAN is statistically superior/inferior to the comparing approaches on each data set with pairwise t-test (at 0.05 significance level).

| Dataset | #label | LIFT | MUENLPLR | SENCE | LIMIC | SLAN |
|---|---|---|---|---|---|---|
| | | *Average precision* (the greater, the better) | | | | |
| llog | 0 | 0.496±0.035 | 0.440±0.023● | **0.506±0.044** | 0.478±0.035 | 0.497±0.039 |
| | 3 | 0.479±0.035 | 0.425±0.026● | **0.487±0.044** | 0.466±0.033 | 0.484±0.038 |
| | 5 | 0.473±0.035 | 0.421±0.025● | **0.480±0.041** | 0.460±0.030 | 0.477±0.036 |
| | 7 | 0.469±0.035 | 0.416±0.022● | **0.478±0.040** | 0.459±0.027 | 0.473±0.035 |
| | 9 | 0.469±0.029 | 0.413±0.021● | **0.476±0.034** | 0.459±0.024 | 0.469±0.032 |
| enron | 0 | 0.727±0.041● | 0.663±0.054● | 0.747±0.037 | 0.695±0.052● | **0.751±0.035** |
| | 6 | 0.674±0.031● | 0.612±0.033● | 0.682±0.032 | 0.653±0.031● | **0.687±0.029** |
| | 9 | 0.659±0.035● | 0.595±0.028● | 0.667±0.034 | 0.641±0.026● | **0.669±0.035** |
| | 12 | 0.655±0.025● | 0.591±0.011● | 0.663±0.023 | 0.638±0.020● | **0.664±0.025** |
| recreation | 0 | 0.685±0.017● | 0.574±0.019● | **0.718±0.027** | 0.632±0.023● | 0.717±0.026 |
| | 4 | 0.661±0.012● | 0.562±0.017● | **0.684±0.025** | 0.619±0.022● | 0.681±0.023 |
| | 6 | 0.659±0.015● | 0.559±0.017● | **0.678±0.022** | 0.618±0.026● | 0.673±0.020 |
| | 8 | 0.653±0.015● | 0.556±0.016● | **0.672±0.021**○ | 0.615±0.025● | 0.666±0.018 |
| slashdot | 0 | 0.563±0.024● | 0.462±0.022● | 0.599±0.026● | 0.542±0.031● | **0.641±0.035** |
| | 3 | 0.553±0.018● | 0.456±0.018● | 0.587±0.020● | 0.534±0.029● | **0.628±0.029** |
| | 5 | 0.541±0.016● | 0.450±0.013● | 0.575±0.015● | 0.520±0.025● | **0.617±0.026** |
| | 7 | 0.541±0.018● | 0.449±0.012● | 0.574±0.017● | 0.518±0.020● | **0.614±0.023** |
| corel5k | 0 | 0.463±0.011● | 0.338±0.012● | **0.480±0.011**○ | 0.385±0.053● | 0.471±0.008 |
| | 7 | 0.426±0.019 | 0.320±0.014● | **0.441±0.022**○ | 0.370±0.057● | 0.431±0.017 |
| | 12 | 0.417±0.014 | 0.316±0.016● | **0.429±0.016**○ | 0.367±0.055● | 0.417±0.014 |
| | 17 | 0.411±0.013 | 0.313±0.011● | **0.423±0.015**○ | 0.368±0.050● | 0.409±0.013 |
| | 22 | 0.404±0.012 | 0.310±0.008● | **0.418±0.014**○ | 0.364±0.050● | 0.404±0.013 |
| arts | 0 | 0.716±0.027● | 0.607±0.036● | 0.749±0.028 | 0.626±0.050● | **0.752±0.023** |
| | 3 | 0.595±0.061● | 0.531±0.044● | 0.613±0.061● | 0.521±0.077● | **0.622±0.059** |
| | 5 | 0.573±0.036● | 0.515±0.029● | 0.588±0.038● | 0.502±0.061● | **0.597±0.035** |
| | 7 | 0.551±0.035● | 0.495±0.023● | 0.566±0.036● | 0.483±0.051● | **0.576±0.034** |
| education | 0 | 0.762±0.029● | 0.684±0.034● | 0.797±0.026● | 0.707±0.045● | **0.805±0.028** |
| | 4 | 0.690±0.056● | 0.637±0.053● | 0.714±0.050● | 0.649±0.068● | **0.722±0.048** |
| | 6 | 0.678±0.036● | 0.625±0.032● | 0.701±0.033● | 0.638±0.043● | **0.709±0.033** |
| rcvsubset2-2 | 0 | 0.660±0.009● | 0.437±0.022● | 0.726±0.011● | 0.465±0.089● | **0.752±0.015** |
| | 8 | 0.525±0.024● | 0.375±0.018● | 0.575±0.029● | 0.418±0.060● | **0.596±0.029** |
| | 12 | 0.504±0.025● | 0.365±0.021● | 0.551±0.027● | 0.411±0.066● | **0.572±0.026** |
| | 16 | 0.485±0.016● | 0.357±0.016● | 0.528±0.019● | 0.400±0.062● | **0.549±0.018** |
| | 20 | 0.473±0.013● | 0.350±0.013● | 0.513±0.017● | 0.393±0.060● | **0.532±0.017** |
| bibtex | 0 | 0.610±0.029● | 0.348±0.011● | 0.600±0.046● | 0.530±0.018● | **0.778±0.031** |
| | 5 | 0.617±0.035● | 0.356±0.013● | 0.613±0.047● | 0.546±0.023● | **0.782±0.031** |
| | 8 | 0.614±0.037● | 0.356±0.014● | 0.609±0.052● | 0.546±0.034● | **0.778±0.033** |
| | 11 | 0.608±0.040● | 0.353±0.011● | 0.603±0.053● | 0.541±0.030● | **0.772±0.037** |
| | 14 | 0.606±0.028● | 0.355±0.009● | 0.601±0.039● | 0.542±0.022● | **0.770±0.030** |

Table 5: Experimental results of each compared approach (mean±std) in terms of *Macro-averaging AUC* with different label batch size (denoted by #label). The best and the second best performance of each data set methods are highlighted in **boldface** and underline respectively. In addition, ●/○ indicates whether SLAN is statistically superior/inferior to the comparing approaches on each data set with pairwise t-test (at 0.05 significance level).

| Dataset | #label | LIFT | MUENLPLR | SENCE | LIMIC | SLAN |
|---|---|---|---|---|---|---|
| | | *Macro-averaging AUC* (the greater, the better) | | | | |
| llog | 0 | 0.590±0.029● | 0.541±0.019● | 0.595±0.031● | 0.512±0.016● | **0.618±0.033** |
| | 3 | 0.579±0.031 | 0.526±0.021● | 0.587±0.034 | 0.511±0.015● | **0.602±0.034** |
| | 5 | 0.573±0.027 | 0.524±0.021● | 0.581±0.026 | 0.509±0.014● | **0.594±0.034** |
| | 7 | 0.573±0.020 | 0.521±0.019● | 0.577±0.022 | 0.509±0.013● | **0.590±0.033** |
| | 9 | 0.570±0.019 | 0.522±0.017● | 0.573±0.023 | 0.508±0.011● | **0.585±0.034** |
| enron | 0 | 0.688±0.023● | 0.589±0.026● | 0.690±0.049● | 0.580±0.021● | **0.725±0.034** |
| | 6 | 0.646±0.026● | 0.565±0.017● | 0.660±0.036● | 0.561±0.016● | **0.681±0.033** |
| | 9 | 0.645±0.024● | 0.563±0.014● | 0.660±0.036 | 0.561±0.015● | **0.679±0.031** |
| | 12 | 0.646±0.022● | 0.563±0.013● | 0.659±0.033 | 0.560±0.014● | **0.677±0.030** |
| recreation | 0 | 0.702±0.022● | 0.585±0.016● | 0.739±0.028● | 0.564±0.020● | **0.758±0.026** |
| | 4 | 0.647±0.014● | 0.561±0.010● | 0.671±0.020● | 0.544±0.010● | **0.687±0.019** |
| | 6 | 0.641±0.011● | 0.557±0.011● | 0.662±0.018● | 0.542±0.010● | **0.679±0.017** |
| | 8 | 0.635±0.010● | 0.555±0.010● | 0.655±0.015● | 0.538±0.010● | **0.670±0.015** |
| slashdot | 0 | 0.606±0.023● | 0.522±0.014● | 0.667±0.025● | 0.564±0.009● | **0.728±0.030** |
| | 3 | 0.601±0.016● | 0.522±0.009● | 0.658±0.020● | 0.556±0.010● | **0.699±0.026** |
| | 5 | 0.596±0.012● | 0.522±0.007● | 0.653±0.014● | 0.553±0.010● | **0.692±0.022** |
| | 7 | 0.596±0.012● | 0.522±0.008● | 0.653±0.016● | 0.552±0.011● | **0.687±0.020** |
| corel5k | 0 | 0.697±0.021● | 0.573±0.009● | 0.705±0.019● | 0.535±0.020● | **0.719±0.016** |
| | 7 | 0.677±0.017● | 0.567±0.009● | 0.686±0.016● | 0.531±0.018● | **0.697±0.013** |
| | 12 | 0.668±0.017● | 0.565±0.010● | 0.676±0.016● | 0.530±0.017● | **0.687±0.014** |
| | 17 | 0.660±0.015● | 0.560±0.009● | 0.670±0.016● | 0.529±0.016● | **0.679±0.014** |
| | 22 | 0.657±0.016● | 0.559±0.008● | 0.667±0.017● | 0.528±0.015● | **0.676±0.015** |
| arts | 0 | 0.731±0.022● | 0.614±0.015● | 0.762±0.020● | 0.585±0.030● | **0.782±0.022** |
| | 3 | 0.652±0.023● | 0.572±0.016● | 0.672±0.022● | 0.554±0.019● | **0.684±0.022** |
| | 5 | 0.638±0.015● | 0.564±0.014● | 0.656±0.015● | 0.549±0.015● | **0.665±0.015** |
| | 7 | 0.630±0.014● | 0.558±0.012● | 0.645±0.014● | 0.545±0.015● | **0.655±0.013** |
| education | 0 | 0.724±0.021● | 0.592±0.028● | 0.777±0.022● | 0.593±0.029● | **0.792±0.022** |
| | 4 | 0.629±0.013● | 0.555±0.014● | 0.667±0.013● | 0.552±0.018● | **0.681±0.015** |
| | 6 | 0.623±0.010● | 0.553±0.014● | 0.660±0.012● | 0.549±0.019● | **0.674±0.016** |
| rcvsubset2-2 | 0 | 0.798±0.015● | 0.596±0.021● | 0.839±0.012● | 0.538±0.027● | **0.864±0.010** |
| | 8 | 0.729±0.022● | 0.568±0.010● | 0.769±0.022● | 0.526±0.020● | **0.789±0.017** |
| | 12 | 0.722±0.019● | 0.563±0.014● | 0.763±0.018● | 0.524±0.017● | **0.782±0.014** |
| | 16 | 0.711±0.015● | 0.559±0.011● | 0.750±0.015● | 0.521±0.015● | **0.770±0.012** |
| | 20 | 0.705±0.015● | 0.555±0.011● | 0.744±0.015● | 0.520±0.015● | **0.764±0.014** |
| bibtex | 0 | 0.718±0.025● | 0.529±0.011● | 0.586±0.047● | 0.593±0.017● | **0.840±0.015** |
| | 5 | 0.709±0.027● | 0.529±0.012● | 0.583±0.046● | 0.591±0.017● | **0.833±0.015** |
| | 8 | 0.706±0.027● | 0.528±0.012● | 0.582±0.045● | 0.590±0.016● | **0.830±0.016** |
| | 11 | 0.704±0.028● | 0.527±0.012● | 0.584±0.044● | 0.589±0.015● | **0.827±0.016** |
| | 14 | 0.703±0.027● | 0.527±0.013● | 0.583±0.045● | 0.588±0.015● | **0.827±0.016** |

Table 6: Win/tie/loss counts (pairwise t-test at 0.05 significant level) for SLAN against other multi-label approaches.

| Metrics | SLAN **against** | | | |
|---|---|---|---|---|
| | LIFT | MUENLPLR | SENCE | LIMIC |
| *Ranking loss* | 27/7/5 | 39/0/0 | 16/15/8 | 29/10/0 |
| *One-error* | 34/5/0 | 39/0/0 | 20/19/0 | 34/5/0 |
| *Coverage* | 28/6/5 | 39/0/0 | 18/15/6 | 29/10/0 |
| *Average precision* | 30/9/0 | 39/0/0 | 20/13/6 | 34/5/0 |
| *Macro-averaging AUC* | 35/4/0 | 39/0/0 | 33/6/0 | 39/0/0 |
| **In Total** | **154/31/10** | **195/0/0** | **107/68/20** | **165/30/0** |

Table 7: Experimental results of each compared approach (mean±std) in terms of *Coverage* with different label batch size (denoted by #label). The best and the second best performance of each data set methods are highlighted in **boldface** and underline respectively. In addition, ●/○ indicates whether SLAN is statistically superior/inferior to the comparing approaches on each data set with pairwise t-test (at 0.05 significance level).

| Dataset | #label | LIFT | MUENLPLR | SENCE | LIMIC | SLAN |
|---|---|---|---|---|---|---|
| | | | *Coverage* (the smaller, the better) | | | |
| llog | 0 | 0.332±0.030 | 0.384±0.027● | **0.329±0.035** | 0.340±0.031 | 0.336±0.028 |
| | 3 | **0.348±0.032** | 0.398±0.027● | **0.348±0.036** | 0.353±0.034 | **0.348±0.029** |
| | 5 | 0.354±0.033 | 0.401±0.026● | **0.351±0.033** | 0.358±0.031 | 0.354±0.030 |
| | 7 | 0.356±0.029 | 0.403±0.022● | **0.355±0.029** | 0.361±0.027 | 0.357±0.027 |
| | 9 | **0.356±0.026** | 0.404±0.020● | **0.356±0.026** | 0.361±0.026 | 0.361±0.026 |
| enron | 0 | 0.419±0.053● | 0.488±0.054● | 0.401±0.051 | 0.442±0.061● | **0.399±0.054** |
| | 6 | 0.375±0.031● | 0.445±0.020● | 0.367±0.025 | 0.391±0.031● | **0.363±0.028** |
| | 9 | 0.368±0.021● | 0.442±0.010● | 0.361±0.016 | 0.381±0.025● | **0.358±0.019** |
| | 12 | 0.366±0.015● | 0.440±0.012● | 0.360±0.009 | 0.380±0.016● | **0.359±0.013** |
| recreation | 0 | 0.282±0.019● | 0.367±0.021● | 0.263±0.024 | 0.313±0.025● | **0.261±0.026** |
| | 4 | 0.303±0.017● | 0.378±0.016● | **0.290±0.023** | 0.322±0.022● | 0.292±0.023 |
| | 6 | 0.304±0.014● | 0.379±0.014● | **0.294±0.020** | 0.322±0.020● | 0.295±0.018 |
| | 8 | 0.306±0.015 | 0.379±0.014● | **0.296±0.018** | 0.320±0.019● | 0.300±0.016 |
| slashdot | 0 | 0.296±0.028● | 0.374±0.024● | 0.264±0.027● | 0.303±0.031● | **0.244±0.029** |
| | 3 | 0.303±0.022● | 0.377±0.018● | 0.273±0.021● | 0.312±0.028● | **0.250±0.023** |
| | 5 | 0.312±0.018● | 0.381±0.012● | 0.281±0.016● | 0.322±0.023● | **0.256±0.020** |
| | 7 | 0.311±0.020● | 0.380±0.011● | 0.281±0.020● | 0.322±0.022● | **0.257±0.018** |
| corel5k | 0 | 0.310±0.012○ | 0.441±0.013● | **0.295±0.010○** | 0.372±0.037 | 0.357±0.009 |
| | 7 | 0.316±0.018○ | 0.430±0.015● | **0.304±0.020○** | 0.375±0.044 | 0.361±0.013 |
| | 12 | 0.322±0.020○ | 0.431±0.019● | **0.313±0.023○** | 0.379±0.048 | 0.368±0.016 |
| | 17 | 0.330±0.022○ | 0.435±0.018● | **0.320±0.025○** | 0.383±0.051 | 0.375±0.016 |
| | 22 | 0.333±0.023○ | 0.434±0.015● | **0.322±0.025○** | 0.384±0.051 | 0.375±0.016 |
| arts | 0 | 0.274±0.033● | 0.351±0.036● | 0.249±0.031● | 0.339±0.048● | **0.242±0.028** |
| | 3 | 0.351±0.045● | 0.398±0.038● | 0.339±0.046● | 0.412±0.065● | **0.325±0.045** |
| | 5 | 0.370±0.027● | 0.411±0.026● | 0.359±0.029● | 0.431±0.064● | **0.343±0.029** |
| | 7 | 0.379±0.030● | 0.420±0.023● | 0.368±0.036● | 0.436±0.059● | **0.351±0.031** |
| education | 0 | 0.256±0.036● | 0.324±0.042● | 0.232±0.029● | 0.297±0.047● | **0.225±0.031** |
| | 4 | 0.311±0.051● | 0.358±0.051● | 0.293±0.045● | 0.346±0.066● | **0.286±0.044** |
| | 6 | 0.318±0.034● | 0.365±0.036● | 0.301±0.030● | 0.355±0.041● | **0.294±0.032** |
| rcvsubset2-2 | 0 | 0.204±0.011● | 0.361±0.013● | **0.171±0.013○** | 0.320±0.039● | 0.175±0.015 |
| | 8 | 0.310±0.038● | 0.419±0.029● | 0.281±0.036 | 0.377±0.044● | **0.277±0.027** |
| | 12 | 0.323±0.033● | 0.425±0.025● | 0.295±0.030 | 0.385±0.047● | **0.288±0.028** |
| | 16 | 0.335±0.022● | 0.428±0.016● | 0.307±0.020 | 0.393±0.034● | **0.300±0.017** |
| | 20 | 0.351±0.021● | 0.439±0.012● | 0.323±0.021 | 0.404±0.032● | **0.313±0.019** |
| bibtex | 0 | 0.240±0.033● | 0.405±0.015● | 0.271±0.056● | 0.286±0.018● | **0.140±0.023** |
| | 5 | 0.236±0.035● | 0.396±0.015● | 0.265±0.053● | 0.281±0.024● | **0.139±0.021** |
| | 8 | 0.237±0.040● | 0.396±0.021● | 0.268±0.056● | 0.282±0.036● | **0.142±0.023** |
| | 11 | 0.244±0.041● | 0.401±0.020● | 0.272±0.056● | 0.290±0.028● | **0.149±0.024** |
| | 14 | 0.247±0.034● | 0.402±0.014● | 0.275±0.047● | 0.293±0.021● | **0.151±0.020** |

Table 8: Win/tie/loss counts (pairwise t-test at 0.05 significant level) for SLAN against other anomaly detection approaches.

| Metrics | SLAN **against** | | |
|---|---|---|---|
| | OC-SVM | IFOREST | MUENLFOREST |
| *F-measure* | 26/4/0 | 26/4/0 | 23/7/0 |

Table 9: Experimental results of each compared approach (mean±std) in terms of *One-error* with different label batch size (denoted by #label). The best and the second best performance of each data set methods are highlighted in **boldface** and underline respectively. In addition, ●/○ indicates whether SLAN is statistically superior/inferior to the comparing approaches on each data set with pairwise t-test (at 0.05 significance level).

| Dataset | #label | LIFT | MUENLPLR | SENCE | LIMIC | SLAN |
|---|---|---|---|---|---|---|
| | | *One-error* (the smaller, the better) | | | | |
| llog | 0 | 0.708±0.040 | 0.766±0.034● | **0.687±0.054** | 0.729±0.041 | 0.702±0.057 |
| | 3 | 0.724±0.039 | 0.781±0.035● | **0.705±0.050** | 0.740±0.036 | 0.714±0.050 |
| | 5 | 0.730±0.039 | 0.785±0.033● | **0.715±0.046** | 0.747±0.033 | 0.722±0.050 |
| | 7 | 0.733±0.040 | 0.794±0.031● | **0.716±0.046** | 0.745±0.026 | 0.725±0.050 |
| | 9 | 0.734±0.035 | 0.797±0.029● | **0.717±0.040** | 0.745±0.022 | 0.729±0.043 |
| enron | 0 | 0.230±0.105● | 0.293±0.156● | 0.210±0.099 | 0.276±0.146● | **0.205±0.093** |
| | 6 | 0.393±0.079● | 0.436±0.090● | 0.376±0.076 | 0.444±0.104● | **0.370±0.069** |
| | 9 | 0.436±0.082● | 0.479±0.077● | 0.416±0.080 | 0.484±0.086● | **0.413±0.076** |
| | 12 | 0.446±0.044● | 0.489±0.024● | 0.426±0.051 | 0.493±0.054● | **0.423±0.048** |
| recreation | 0 | 0.469±0.031● | 0.612±0.030● | 0.413±0.039 | 0.577±0.035● | **0.411±0.036** |
| | 4 | 0.501±0.020● | 0.629±0.027● | **0.459±0.033** | 0.592±0.029● | 0.460±0.032 |
| | 6 | 0.503±0.026● | 0.633±0.024● | **0.467±0.032** | 0.592±0.040● | 0.470±0.030 |
| | 8 | 0.512±0.027● | 0.636±0.023● | **0.475±0.030** | 0.599±0.039● | 0.479±0.028 |
| slashdot | 0 | 0.632±0.033● | 0.762±0.028● | 0.587±0.031● | 0.668±0.040● | **0.524±0.047** |
| | 3 | 0.642±0.033● | 0.768±0.024● | 0.601±0.027● | 0.674±0.038● | **0.543±0.039** |
| | 5 | 0.657±0.032● | 0.774±0.021● | 0.617±0.023● | 0.691±0.035● | **0.557±0.036** |
| | 7 | 0.657±0.033● | 0.775±0.020● | 0.618±0.024● | 0.694±0.028● | **0.562±0.033** |
| corel5k | 0 | 0.652±0.014● | 0.781±0.014● | 0.634±0.017 | 0.732±0.024● | **0.628±0.014** |
| | 7 | 0.706±0.018● | 0.808±0.026● | 0.687±0.026● | 0.756±0.030● | **0.678±0.025** |
| | 12 | 0.719±0.012● | 0.809±0.016● | 0.705±0.018● | 0.758±0.025● | **0.694±0.017** |
| | 17 | 0.725±0.011● | 0.812±0.011● | 0.711±0.014● | 0.755±0.017● | **0.702±0.013** |
| | 22 | 0.735±0.011● | 0.817±0.009● | 0.718±0.011● | 0.765±0.013● | **0.709±0.012** |
| arts | 0 | 0.393±0.035● | 0.550±0.049● | **0.345±0.038** | 0.531±0.064● | 0.347±0.032 |
| | 3 | 0.556±0.076● | 0.648±0.050● | 0.532±0.079 | 0.659±0.088● | **0.528±0.076** |
| | 5 | 0.583±0.043● | 0.666±0.034● | 0.563±0.048 | 0.681±0.064● | **0.560±0.043** |
| | 7 | 0.616±0.041● | 0.692±0.026● | 0.595±0.040 | 0.707±0.053● | **0.592±0.035** |
| education | 0 | 0.372±0.043● | 0.482±0.049● | 0.313±0.041 | 0.469±0.067● | **0.301±0.040** |
| | 4 | 0.480±0.077● | 0.549±0.071● | 0.440±0.069● | 0.544±0.085● | **0.428±0.067** |
| | 6 | 0.498±0.049● | 0.568±0.042● | 0.460±0.046● | 0.557±0.056● | **0.449±0.046** |
| rcvsubset2-2 | 0 | 0.435±0.016● | 0.694±0.036● | 0.350±0.014● | 0.686±0.091● | **0.311±0.018** |
| | 8 | 0.585±0.018● | 0.758±0.023● | 0.516±0.024● | 0.731±0.081● | **0.482±0.028** |
| | 12 | 0.609±0.021● | 0.767±0.025● | 0.544±0.024● | 0.739±0.077● | **0.511±0.025** |
| | 16 | 0.632±0.017● | 0.776±0.023● | 0.571±0.019● | 0.752±0.074● | **0.537±0.021** |
| | 20 | 0.640±0.012● | 0.781±0.017● | 0.584±0.016● | 0.753±0.071● | **0.555±0.014** |
| bibtex | 0 | 0.498±0.029● | 0.832±0.016● | 0.486±0.043● | 0.611±0.037● | **0.281±0.039** |
| | 5 | 0.489±0.039● | 0.824±0.018● | 0.467±0.047● | 0.589±0.033● | **0.274±0.039** |
| | 8 | 0.491±0.038● | 0.824±0.016● | 0.472±0.054● | 0.585±0.037● | **0.279±0.042** |
| | 11 | 0.499±0.039● | 0.827±0.011● | 0.481±0.056● | 0.592±0.032● | **0.286±0.046** |
| | 14 | 0.500±0.024● | 0.824±0.013● | 0.482±0.036● | 0.589±0.025● | **0.287±0.037** |

## A.2 Data and Code Availability

Our source code can be found at `https://palm.seu.edu.cn/zhangml/`. The datasets used in this paper are public, and can be found in Section 4.1.

