# OpenReview forum: "Multi-Label Open Set Recognition"
_NeurIPS.cc/2024/Conference — NeurIPS 2024 poster_

### Official Review · Reviewer_wpso · 2024-07-01

**Soundness:** 3
**Presentation:** 3
**Contribution:** 2
**Rating:** 5
**Confidence:** 4

**Summary:**

This paper addresses the problem of classifying instances with unknown labels in a multi-label setting (multi-label open set recognition). A
novel approach named SLAN is proposed which leverages sub-labeling information, and unknown labels are recognized by differentiating the sub-labeling information from holistic supervision. The effectiveness of the proposed method is validated on different benchmarks.

**Strengths:**

- The paper presents a detailed and well-formulated methodology, including the method for structural information discovery and the part related to optimization.

- Extensive experiments on various datasets are conducted to prove the effectiveness of the proposed SLAN approach. The empirical results demonstrate superior or comparable performance to the mentioned methods.

**Weaknesses:**

- The paper introduces the MLOSR problem, which is novel, but it may overlap with existing tasks like zero-shot multi-label classification and open-vocabulary multi-label classification.

- This paper has limited baselines for comparison. While the paper includes comparisons with several multi-label learning and anomaly detection approaches, it should also compare with state-of-the-art methods specifically designed for zero-shot and open-vocabulary multi-label classification.

- The proposed SLAN approach need to solve multiple optimization problems, which may not scale well to large datasets.

**Questions:**

- There are existing tasks with similar settings. For example, in image tasks, there are zero-shot multi-label classification, and open vocabulary multi-label classification. What's the difference between the proposed task and these tasks? Is the setting more applicable?

- It would be better if the paper can include state-of-the-art methods for zero-shot and open-vocabulary multi-label classification in the experimental comparison.

- It would be better to include a flowchart outlining the key step in the proposed approach, which can improve the clarity and make it easier to follow.

**Limitations:**

The limitation and societal impact is addressed.

---

> ### Author Rebuttal · Authors · 2024-08-07
>
> 1. There are existing tasks with similar settings. For example, in image tasks, there are zero-shot multi-label classification, and open vocabulary multi-label classification. What's the difference between the proposed task and these tasks? Is the setting more applicable?
>
> - **Response 1:** Thanks to the comment. Although multi-label open set recognition (MLOSR), multi-label zero-shot classification (MLZSL) and open vocabulary multi-label classification (OVML) share similar goals, there are essential differences between them. During training phase, in addition to the training set, MLZSL and OVML needs additional prior knowledge of unseen labels ,whereas MLOSR does not. Specifically, MLOSR solely trains on the training set consist of instance-labels pairs. MLZSL additionally needs to extract the semantic information from the observed label space, that is, the relevant label embeddings and the relation between seen and unseen label embeddings. In OVML, visual-related language data like image captions can serve as auxiliary supervision , which potentially has an implicit intersection with the unseen labels. During testing, due to the lack of prior knowledge, MLOSR only needs to identify novel labels and mark them as unknown, while MLZSL and OVML must classify unknown labels into specific labels.
>
>   MLOSR can be considered as an extreme case of MLZSL and OVML. Each of these tasks has its own applicable real-world scenarios. It is necessary to determine the specific task based on the availability of prior knowledge.
>
> ---
>
> 2. It would be better if the paper can include state-of-the-art methods for zero-shot and open-vocabulary multi-label classification in the experimental comparison.
>
> - **Response 2:**  Thanks to the suggestion. As mentioned above, MLOSR is differentiated from MLZSL and OVML based on whether there is prior knowledge of unseen labels or not. Thus, it is inappropriate to apply approaches designed for MLOSR to MLZSL and OVML scenarios without requisite prior knowledge, and vice versa.
>
> ---
>
> 3. It would be better to include a flowchart outlining the key step in the proposed approach, which can improve the clarity and make it easier to follow.
>
> - **Response 3:** Thanks to the suggestion. The flowchart of SLAN is summarized in Figure 1 in the Rebuttal PDF file. Given the multi-label training set, a weighted graph is constructed to characterize the manifold structure of feature space. After that, the alternative optimization strategy is adopted to optimize open set recognizer and multi-label classifier simultaneously.
> ---
>
> 4. The paper introduces the MLOSR problem, which is novel, but it may overlap with existing tasks like zero-shot multi-label classification and open-vocabulary multi-label classification.
>
> - **Response 4:** Please kindly refer to **Response 1**.
> ---
>
> 5. This paper has limited baselines for comparison. While the paper includes comparisons with several multi-label learning and anomaly detection approaches, it should also compare with state-of-the-art methods specifically designed for zero-shot and open-vocabulary multi-label classification.
>
> - **Response 5:** Please kindly refer to **Response 2**.
>
> ---
>
> 6. The proposed SLAN approach need to solve multiple optimization problems, which may not scale well to large datasets.
>
> - **Response 6:** Let $q$, $d$ and $m$ denote the number of labels, number of training instances and dimension of input space. Our optimization algorithm mainly contains several steps. Before alternative optimization, we need to instantiate $\mathbf{S}$ with ADMM in $\mathcal{O}(m^3)$. Then our method needs to iteratively solve three optimization problem w.r.t. $\mathbf{Z},\mathbf{F}_k,\mathbf{W}$, which can be solved in $\mathcal{O}(qmd),\mathcal{O}(qm^2)$ and $\mathcal{O}(m^2d)$  respectively. In summary, the overall complexity of our optimization algorithm is the sum of these operations mentioned above. It is noteworthy that the complexity of SLAN is related to $q$ and $m^3$ , which may be slow when applied to date sets with a large number of labels and instances. We will leave efficiency improvement for future work.

---

> > ### Author Response · Authors · 2024-08-11
> > **Looking forward to your feedback**
> >
> > Dear Reviewer wpso, thanks for your thoughtful comments. We believe we have addressed your concerns in our response, including clarifying the difference between the proposed task and zero-shot multi-label classification/open vocabulary multi-label classification, and providing a flowchart for the proposed approach. We would appreciate your thoughts on our response. If you have any remaining or further questions for us to address, we are keen to take the opportunity to do so before the discussion period closes. Thank you!

---

> > ### Comment · Reviewer_wpso · 2024-08-11
> >
> > The author has addressed most of my concerns and the score is updated.

---

> > > ### Author Response · Authors · 2024-08-12
> > > **Thanks**
> > >
> > > Thank you again for the valuable comments. We will check the manuscript again and add the discussion in the revised version.

---

### Official Review · Reviewer_vnjs · 2024-07-09

**Soundness:** 3
**Presentation:** 4
**Contribution:** 3
**Rating:** 7
**Confidence:** 5

**Summary:**

This article introduces a new problem in multi-label open set recognition (MLOSR) and proposes a novel approach named Sub-Labeling Information Reconstruction for MLOSR (SLAN). SLAN utilizes sub-labeling information enriched by structural details in the feature space. Experimental results across various datasets demonstrate its effectiveness in addressing this new challenge of MLOSR.

**Strengths:**

1. This article focuses on a novel problem within multi-label open set recognition (MLOSR), expanding beyond traditional multi-label learning by tackling the identification of unknown open labels. This problem is particularly relevant in practical applications due to the dynamic and open nature of real-world environments.
2. The strength of the paper lies in its clear and thorough analysis and description of the SLAN method.
3. The analysis of SLAN across various experimental metrics on different datasets is comprehensive, providing a robust demonstration of its effectiveness in the experiments.
4. The paper's process and structure are concise, making it relatively easy to understand.

**Weaknesses:**

1. In the Parameter Sensitivity Analysis section, there is some explanation regarding the experimental parameter settings, but it primarily involves varying each parameter individually without further elaboration on their combined effects or interactions. It would be beneficial to include more extensive experimental analysis that considers the interactions between different parameter combinations.
2. The paper asserts that OC-SVM, IFOREST, and MUENLFOREST do not exhibit performance on par with SLAN. It would be advantageous to provide further elaboration on why the parameters of these comparison methods, particularly MUENLFOREST, are configured as they are.
3. Is there any insight into how SLAN performs differently compared to other methods on different datasets? Could this difference be related to dataset characteristics and distributions?

**Questions:**

1. There is concern regarding the robustness of the method. How stable is its performance when errors or irrelevant labels appear in the dataset?

---

> ### Author Rebuttal · Authors · 2024-08-07
>
> 1. There is concern regarding the robustness of the method. How stable is its performance when errors or irrelevant labels appear in the dataset?
>
> - **Response 1:** Thanks to the comment. The detailed experimental results in terms of _ranking loss_ and _F-measure_ under different error rates are reported in the follwing table. As mitigating errors is not considered in SLAN, the performance of SLAN exhibits a slight degradation when the error rate increases. For multi-label classification, the presence of errors or irrelevant labels will hinder the multi-label classifier to accurately induce decision boundary. Similarly, for open-set recognition, such errors or irrelevant labels will impair the discrimination between sub-labeling information and labeling information with holistic supervision. For future work, identifying and mitigating these errors or irrelevant labels will be considered.
> |   |    |   | Ranking loss |  |    |    |   | F-measure |  |
> |:-:|:-:|:-:|:-:|:-:|:-:|:-:|:-:|:-:|:-:|
> |   |   |   | Error rate |  |    |   |   | Error rate |  |
> |Dataset| #label |  0 | 0.1 | 0.3 | Dataset| #label |  0 | 0.1 | 0.3 |
> |enron| 6 | 0.169$\pm$0.011 | 0.182$\pm$0.012 | 0.208$\pm$0.016 |enron| 6  | 0.406$\pm$0.096  | 0.381$\pm$0.085  | 0.375$\pm$0.087  |
> || 9 | 0.172$\pm$0.012 | 0.186$\pm$0.015 | 0.213$\pm$0.019 |  | 9  | 0.321$\pm$0.095  | 0.300$\pm$0.081  | 0.294$\pm$0.079  |
> || 12 | 0.174$\pm$0.010 | 0.188$\pm$0.013 | 0.214$\pm$0.016 | | 12  | 0.281$\pm$0.098  | 0.262$\pm$0.084  | 0.256$\pm$0.081  |
> |slashdot|  3 | 0.260$\pm$0.022 | 0.296$\pm$0.022 | 0.340$\pm$0.021 |  slashdot | 3  | 0.528$\pm$0.127  | 0.526$\pm$0.123  | 0.516$\pm$0.119  |
> || 5 | 0.268$\pm$0.019 | 0.304$\pm$0.018 | 0.348$\pm$0.020 |  | 5  | 0.428$\pm$0.105  | 0.427$\pm$0.104  | 0.419$\pm$0.101  |
> ||  7 | 0.270$\pm$0.018 | 0.305$\pm$0.015 | 0.350$\pm$0.016 | | 7  | 0.350$\pm$0.069  | 0.350$\pm$0.068  | 0.344$\pm$0.067  |
> |corel5k| 7 | 0.266$\pm$0.013 | 0.310$\pm$0.013 | 0.366$\pm$0.011 | corel5k | 7  | 0.653$\pm$0.015  | 0.635$\pm$0.012  | 0.568$\pm$0.013  |
> ||  12 | 0.276$\pm$0.012 | 0.318$\pm$0.011 | 0.373$\pm$0.010 |  | 12  | 0.539$\pm$0.017  | 0.527$\pm$0.015  | 0.482$\pm$0.016  |
> ||  17 | 0.283$\pm$0.012 | 0.324$\pm$0.010 | 0.376$\pm$0.009 |  | 17  | 0.462$\pm$0.012  | 0.454$\pm$0.013  | 0.420$\pm$0.014  |
> ||  22 | 0.286$\pm$0.013 | 0.327$\pm$0.009 | 0.378$\pm$0.008 | | 22  | 0.414$\pm$0.015  | 0.408$\pm$0.016  | 0.381$\pm$0.016  |
>
> ---
>
> 2. In the Parameter Sensitivity Analysis section, there is some explanation regarding the experimental parameter settings, but it primarily involves varying each parameter individually without further elaboration on their combined effects or interactions. It would be beneficial to include more extensive experimental analysis that considers the interactions between different parameter combinations.
>
> - **Response 2:** Thanks to the suggestion. The following table illustrates how the performance of SLAN changes with varying $\gamma$ and $\beta$ on data set enron. SLAN achieves relatively stable performance on _ranking loss_ and somewhat sensitive on _F-measure_, whose trend is similar to when only one parameter is changed. We will elaborate on more different parameter combinations in the revised version.
> ||||  Ranking  |  loss ||| |  |   |    |  F-measure |  |  |
> |:-:|:-:|:-:|:-:|:-:|:-:|:-:|:-:|:-:|:-:|:-:|:-:|:-:|:-:|
> |   |   |   |   | $\beta$  |  |  |   |   |   |   | $\beta$  |  |  |
> ||| 0.001 | 0.01  | 0.1   | 1     | 10     |    |    | 0.001 | 0.01  | 0.1   | 1     | 10     |
> || 0.001 | 0.1734| 0.1730| 0.1698| 0.1802| 0.1908 |     | 0.001 | 0.1701 | 0.0618 | 0.1300 | 0.0749 | 0.0859 |
> || 0.01  | 0.1736| 0.1724| 0.1701| 0.1757| 0.1828 |      | 0.01  | 0.1808 | 0.0964 | 0.1771 | 0.0742 | 0.0820 |
> | $\gamma$   | 0.1   | 0.1707| 0.1703| 0.1686| 0.1674| 0.1684 |  $\gamma$    | 0.1   | 0.2989 | 0.2783 | 0.2408 | 0.2094 | 0.2248 |
> |    | 1     | 0.1726| 0.1726| 0.1719| 0.1714| 0.1711 |      | 1     | 0.3033 | 0.3011 | 0.2812 | 0.2636 | 0.2472 |
> |    | 10    | 0.1742| 0.1741| 0.1741| 0.1740| 0.1740 |      | 10    | 0.3030 | 0.3023 | 0.2929 | 0.2313 | 0.2113 |
>
> ---
>
> 3.  The paper asserts that OC-SVM, IFOREST, and MUENLFOREST do not exhibit performance on par with SLAN. It would be advantageous to provide further elaboration on why the parameters of these comparison methods, particularly MUENLFOREST, are configured as they are.
>
> - **Response 3:**  Thanks to the suggestion. For OC-SVM, IFOREST, and MUENLFOREST, parameter configurations suggested in respective literatures [1,2] are employed. The following table illustrates how the performance of MUENLFOREST changes with varying parameter configurations on data set enron. The results show that the MUENLFOREST approach is not very sensitive to the settings of parameters.
> ||||||
> |:-:|:-:|:-:|:-:|:-:|
> |$q=$|1|3|5|7|
> |F-measure|0.3158|0.3137|0.3179|0.3165|
> |$e_m=$|7|8|9|10|
> |F-measure|0.3178|0.3151|0.3124|0.3110|
> |$C_1=$|0.001|0.01|0.1|1|
> |F-measure|0.3130|0.3135|0.3143|0.3162|
> |$C_2=$|0.001|0.01|0.1|1|
> |F-measure|0.3115|0.3188|0.3154	|0.3103|
>
>   [1] X.-R. Yu, D.-B. Wang, M.-L. Zhang. Partial label learning with emerging new labels. Machine Learning, 2024, 113(4): 1549-1565.
>
>   [2] Y. Zhu, K.-M. Ting, Z.-H Zhou. Multi-label learning with emerging new labels.  IEEE Transactions on Knowledge and Data Engineering, 2018, 30(10):1901–1914.
>
> ---
>
> 4. Is there any insight into how SLAN performs differently compared to other methods on different datasets? Could this difference be related to dataset characteristics and distributions?
>
> - **Response 4:**  Thanks to the comments. For multi-label classification and open-set recognition, SLAN achieves superior or at least comparable performance against the comparing approaches in most cases. Generally speaking, the increase in number of labels and label density helps differentiating the sub-labeling information from holistic supervision, facilitating the learning process of classifier and recognizer.

---

> > ### Author Response · Authors · 2024-08-11
> > **Looking forward to your feedback**
> >
> > Dear Reviewer vnjs, thanks for your thoughtful comments. We believe we have addressed your concerns in our response. We would appreciate your thoughts on our response. Please let us know if there are any further suggestions on how we can improve to address your comments effectively.

---

### Official Review · Reviewer_zY3w · 2024-07-16

**Soundness:** 3
**Presentation:** 3
**Contribution:** 3
**Rating:** 6
**Confidence:** 4

**Summary:**

The abstract discusses multi-label learning where instances can have multiple labels simultaneously. Traditional approaches assume a closed set scenario where test data labels are predefined during training. However, in real-world situations, new labels can emerge during testing, creating an open and dynamic environment. The paper explores Multi-Label Open Set Recognition (MLOSR), which tackles the challenge of recognizing instances with unknown labels in multi-label settings. The proposed approach, SLAN, utilizes sub-labeling information enriched by structural features in the data space to predict unknown labels. Experimental results across different datasets demonstrate the effectiveness of SLAN in addressing the MLOSR problem.

**Strengths:**

1. The paper is well-written and easy to follow.  It shows comprehensive analyses of the motivation of the studied problem and the proposed method.
2.  The paper introduces a novel learning framework called Multi-Label Open Set Recognition (MLOSR), which provides a new direction for multi-label learning research. To solve the MLOSR problem, an approach named SLAN that utilizes sub-labeling information and holistic supervision to recognize unknown labels is proposed. The techniques of SLAN are sound and well-motivated,  offering an effective solution for open set recognition in a multi-label environment.
3. The paper conducts comprehensive empirical validations on various datasets, showing that the proposed method achieves superior or at least comparable performance across multiple evaluation metrics， which enhances the credibility of the research. The paper includes ah findings and indicates its robustness in multi-label learning.  Extensive sensitivity analysis of the trade-off parameters in the SLAN algorithm are conducted, guiding parameter selection in practical applications.

**Weaknesses:**

Generalization Ability: The SLAN algorithm may have limited generalization ability when dealing with extremely multi-label datasets, restricting its application in broader scenarios.
Feature Representation Limitations: Operating within the multi-label learning framework, SLAN might be constrained by less informative or discriminative feature representations.
Computational Resources: While the paper mentions the computational resources for the experiments, it does not elaborate on the computational efficiency and scalability of the algorithm, which may impact its application to large-scale datasets.
Theoretical Foundation: The paper does not provide theoretical results or proofs, which might reduce the depth of understanding of the algorithm's performance.
Access to Code and Datasets: Although the datasets are public, the paper states that the code will be released after acceptance, which may temporarily limit the reproducibility of the results.

**Questions:**

How does the SLAN algorithm perform on datasets with highly imbalanced label distributions?
Does the parameter sensitivity analysis mentioned in the paper consider the characteristics of different types of datasets?
How adaptable and efficient is the SLAN algorithm for real-time or dynamically changing data streams in practical applications?
Does the paper consider the interpretability of the algorithm, i.e., how to explain the predictive results of SLAN?
For future work, are the authors planning to combine the SLAN algorithm with deep learning models to enhance the quality of feature representation and the performance of the algorithm?

**Limitations:**

Yes

---

> ### Author Rebuttal · Authors · 2024-08-07
>
> 1. How does the SLAN algorithm perform on datasets with highly imbalanced label distributions?
>
> - **Response 1:** Thanks to the comment. The following table summarizes the level of class-imbalance on data sets employed in the experiments including the minimum, maximum and average imbalance ratio across the label space, which shows these data sets are highly imbalanced. Macro-averaging AUC is the mostly-used evaluation metric under class-imbalance scenarios. Table 4 in appendix in terms of macro-averaging AUC shows that SLAN achieves superior or at least comparable performance against the comparing approaches.
> |Dataset|min|max|avg|
> |---|---|---|---|
>   |  llog | 6.064 | 37.968 | 20.826 |
>   |  enron | 1.009 | 43.789 | 16.148 |
>   |  recreation | 4.107 | 47.077 | 14.308 |
>   |  slashdot | 5.265 | 34.524 | 15.131 |
>   |  corel5k | 3.464 | 49.000 | 29.401 |
>   |  arts | 3.055 | 45.296 | 12.770 |
>   |  education | 2.173 | 41.017 | 12.109 |
>   |  rcvsubset2-2 | 3.216 | 47.780 | 26.370 |
>   |  bibtex | 6.097 | 49.306 | 32.877 |
>
> ---
>
> 2. Does the parameter sensitivity analysis mentioned in the paper consider the characteristics of different types of datasets?
>
> - **Response 2:**  Thanks to the comment. The following table illustrates how the performance of SLAN changes with varying parameter configurations on data set llog. The performance of SLAN on both llog and enron exhibits similar variation trends. Thus, for fair comparison, we employ the default parameter setting across all data sets.
> | | | | | | | |
> |---|---|---|---|---|---|---|
> |$\alpha=$ | 0.001 | 0.01 | 0.1 | 1 | 10 |100|
> |_Ranking loss_ | 0.3600  |  0.3600  |  0.3601  |  0.3590  |  0.3596  |  0.3599|
> |_F-measure_ |0.5976  |  0.6106  |  0.6855  |  0.6725  |  0.6411  |  0.6823|
> |$\beta=$ | 0.001 | 0.01 | 0.1 | 1 | 10 |100|
> |_Ranking loss_ | 0.3599  |  0.3598   | 0.3602  |  0.3590   | 0.3569 |   0.3568|
> |_F-measure_  |  0.6859  |  0.6859  |  0.6859  |  0.6725 |   0.3549|    0.2712 |
> |$\gamma=$ | 0.001 | 0.01 | 0.1 | 1 | 10 |100|
> |_Ranking loss_ | 0.3489   | 0.3461 |   0.3437  |  0.3490 |   0.3590  |  0.3600|
> |_F-measure_  |   0.1202   | 0.2554   | 0.1767 |   0.6443 |   0.6725 |   0.6859|
>
> ---
>
> 3. How adaptable and efficient is the SLAN algorithm for real-time or dynamically changing data streams in practical applications?
>
> - **Response 3:** Thanks to the comment. In order to achieve a faster convergence and a good result, we adopt a warm start strategy.
>
>   (1) When an unseen instance $\boldsymbol{x}_u$ appears, parameters can be updated through Algorithm 1 in the Rebuttal PDF file.
>
>   (2) When the prior knowledge indicates that the newly appeared instances associated with the same unknown label, these instances can be added to buffer $B$. Once buffer $B$ is full, a pseudo label assignment $\boldsymbol{p}$ for the unknown label $l_{q+1}$ will be induced, which denotes whether an instance is associated with such unknown label. A new classifier for $l_{q+1}$ will be constructed, and parameters will be updated through Algorithm 2 in the Rebuttal PDF file.
>
>   (3) These updated or augmented parameters can all be considered as a warm start for the subsequent joint optimization procedure, rather than a complete retraining.
>
> ---
> 4. Does the paper consider the interpretability of the algorithm, i.e., how to explain the predictive results of SLAN?
>
> - **Response 4:** Thanks to the comment. Intuitively, for multi-label classifier, the learned weights $W$ transparently indicate the importance of each input variable for each class label. Additionally, to achieve better performance of the predictive model, a kernel extension is further utilized for the general nonlinear case. For open set recognizer, the structural information might not be maintained in the sub-label space. The enriched sub-labeling information for an instance with unknown labels is differentiated from labeling information with holistic supervision. Such difference can serve as a criterion to distinguish whether an instance is associated with unknown class labels. In the future, we will conduct further interpretability study for the proposed method.
>
> ---
>
> 5. For future work, are the authors planning to combine the SLAN algorithm with deep learning models to enhance the quality of feature representation and the performance of the algorithm?
>
> - **Response 5:** Thanks to the comment. In the future, it is promising to design MLOSR approaches with deep learning models. For example, label-specific features will be extracted to for informative and discriminative feature representations generation.
>
> ---
>
> 6. Generalization Ability: The SLAN algorithm may have limited generalization ability when dealing with extremely multi-label datasets, restricting its application in broader scenarios.
>
> - **Response 6:** Thanks to the comment. Let $q$, $d$ and $m$ denote the number of labels, number of training instances and dimension of input space. The training complexity of one iteration in alternative optimization is $\mathcal{O}(qm^2+qmd+m^2d)$. It is noteworthy that SLAN learns multiple sub-labeling information matrices of which the number equals $q$, which may be slow when applied to date sets with a large number of labels. This is inevitable if considering sub-labeling information. We will leave efficiency improvement for future work.
>
> ---
>
> 7. Theoretical Foundation: The paper does not provide theoretical results or proofs, which might reduce the depth of understanding of the algorithm's performance.
>
> - **Response 7:** Thanks to the suggestion. In the future, we will perform detailed theoretical analyses for the proposed method.
>
> ---
>
> 8. Access to Code and Datasets: Although the datasets are public, the paper states that the code will be released after acceptance, which may temporarily limit the reproducibility of the results.
>
> - **Response 8:** Thanks to the comment. The code for this paper will be released  and the results can be reproduced after the paper is accepted.

---

> ### Author Response · Authors · 2024-08-11
> **Looking forward to your feedback**
>
> Dear Reviewer zY3w, thanks for your thoughtful comments. Following your suggestions, we conducted additional experiments regarding parameter sensitivity analysis across different datasets. We would appreciate your thoughts on our response. Please let us know if there is anything else we can do to address your comments.

---

### Author Rebuttal · Authors · 2024-08-07

Dear Reviewers,

We greatly appreciate all of you for your thoughtful comments and valuable suggestions. These are very helpful for improving our paper. We have carefully referred to the questions and written the response. In addition to the text responses, we also report some figure results in the PDF file. We hope the responses would meet your requirements.

Best Regards,

Authors

---

### Decision · Program_Chairs · 2024-09-25

**Decision:**

Accept (poster)

**Comment:**

This paper was reviewed by three experts in the field and received "Weak Accept", "Accept" and "Borderline Accept" as the ratings. The reviewers agreed that the proposed multi-label open set recognition framework introduces a new direction for multi-label learning research, the experimental analysis is comprehensive with encouraging results, and that the paper is well-written and easy to follow.

The reviewers raised concerns about the robustness of the proposed method, which was addressed convincingly by the authors in the rebuttal. The authors also conducted experiments to study the interactions between different parameter combinations, in response to a comment from one of the reviewers. Concerns were also raised about the parameter sensitivity analysis, which was also addressed convincingly by the authors in the rebuttal.

The reviewers, in general, have a positive opinion about the paper and its contributions. Based on the reviewers' feedback, the decision is to recommend the paper for acceptance to NeurIPS 2024. The reviewers have provided some valuable comments, such as the theoretical analyses of the proposed method, its computational efficiency when dealing with datasets with a large number of labels and designing multi-label open set recognition algorithms with deep learning models. The authors are encouraged to address these in the final version of their paper. The authors are also encouraged to make their code publicly available for better reproducibility. We congratulate the authors on the acceptance of their paper!